# Single cell transcriptomics reveals dysregulated cellular and molecular networks in a fragile X syndrome model

Elisa Donnard[1‡]*, Huan Shu[2‡]*, Manuel Garber[1,2,3]*

**1** Program in Bioinformatics and Integrative Biology, University of Massachusetts Medical School, Worcester, Massachusetts, United States of America, **2** Program in Molecular Medicine, University of Massachusetts Medical School, Worcester, Massachusetts, United States of America, **3** Department of Dermatology, Department of Medicine, University of Massachusetts Medical School, Worcester, Massachusetts, United States of America

‡ These authors share first authorship on this work.
* edonnard@broadinstitute.org (ED); dr.huan.shu@gmail.com (HS); manuel.garber@umassmed.edu (MG)

**Data Availability Statement:** Raw and processed data for this study are publicly available at GEO, Accession GSE147191.

**Funding:** This work was partially supported by the National Institutes of Health under award number

## Abstract

Despite advances in understanding the pathophysiology of Fragile X syndrome (FXS), its molecular basis is still poorly understood. Whole brain tissue expression profiles have proved surprisingly uninformative, therefore we applied single cell RNA sequencing to profile an FMRP deficient mouse model with higher resolution. We found that the absence of FMRP results in highly cell type specific gene expression changes that are strongest among specific neuronal types, where FMRP-bound mRNAs were prominently downregulated. Metabolic pathways including translation and respiration are significantly upregulated across most cell types with the notable exception of excitatory neurons. These effects point to a potential difference in the activity of mTOR pathways, and together with other dysregulated pathways, suggest an excitatory-inhibitory imbalance in the *Fmr1*-knock out cortex that is exacerbated by astrocytes. Our data demonstrate that FMRP loss affects abundance of key cellular communication genes that potentially affect neuronal synapses and provide a resource for interrogating the biological basis of this disorder.

## Author summary

Fragile X syndrome is a leading genetic cause of inherited intellectual disability and autism spectrum disorder. It results from the inactivation of a single gene, *FMR1* and hence the loss of its encoded protein FMRP. Despite decades of intensive research, we still lack an overview of the molecular and biological consequences of the disease. Using single cell RNA sequencing, we profiled cells from the brain of healthy mice and of knock-out mice lacking the FMRP protein, a common model for this disease, to identify molecular changes that happen across different cell types. We find neurons are the most impacted cell type, where genes in multiple pathways are similarly impacted. This includes transcripts known to be bound by FMRP, which are collectively decreased only in neurons but not in other cell types. Our results show how the loss of FMRP affects the intricate

U54HD082013 (M.G.). The funders had no role in study design, data collection and analysis, decision to publish, or preparation of the manuscript.

**Competing interests:** The authors have declared that no competing interests exist.

interactions between different brain cell types, which could provide new perspectives to the development of therapeutic interventions.

## Introduction

Fragile X syndrome (FXS) is the most common inherited form of intellectual disability and autism spectrum disorder (ASD). The disease results from the silencing of a single gene (*FMR1*, FMRP Translational Regulator 1), which encodes the RNA-binding protein FMRP [1]. The loss of FMRP leads to a neurodevelopmental disorder with an array of well characterized behaviour and cellular abnormalities, such as impaired cognitive functions, repetitive behaviours, altered synaptic morphology and function [2]; many of which are reproduced in *Fmr1*-knock out (*Fmr1*-KO) mouse models [3].

The molecular pathophysiology of FXS and FMRP function has been the subject of numerous studies over the past decades [4,5]. The most extensively studied function of FMRP is its role as a translational repressor. FMRP is critical to hippocampal long-term synaptic and spine morphological plasticity, dependent on protein synthesis. More specifically, the absence of FMRP leads to an exaggerated long-term synaptic depression, induced by the metabotropic glutamate receptor 5 (mGLUR5-LTD) [6]. However, several ambitious clinical trials that aimed to suppress translation or inhibit mGluR pathways have thus far failed [7]. The significance of FMRP's role as a translation repressor at synapses is not without challenges. First, several teams show evidence that FRMP can function as a translation activator [8–10]. Secondly, only a few mRNAs that are bound by FMRP showed a consistent increase in protein levels upon loss of FMRP, and increased levels of proteins are not always pathogenic [11–15]. More importantly, focusing on FMRP's translational function in dendritic synapses overlooks the fact that the great majority of this protein is located in the cell soma [16]. Alternatively, FMRP could have important functions independent of its role in translation regulation. Indeed, a wide range of research has associated FMRP to multiple steps of the mRNA life cycle, including pre-mRNA splicing [17], mRNA editing [18,19], miRNA activity [20,21], and mRNA stability [22,23]. Additionally, FMRP may function outside the RNA-binding scope, by chromatin binding and regulating genome stability [24,25], as well as directly binding to and regulating ion channels [26,27].

Most of the above-mentioned studies focus on FMRP's function in neurons, and rightly so, as neurons have the highest FMRP protein levels in the brain [16]. Evidence from clinical studies with FXS patients and from *Fmr1*-KO mouse models of the disease supports the view that neurons are the main affected cell type [28,29]. However, other cell types in the brain also express FMRP and FMRP loss has a clear effect on them. Indeed, astrocytes, oligodendrocyte precursor cells, and microglia express FMRP in a brain region and development-dependent manner [30]. FMRP-depleted astrocytes are more reactive [31], and this response alone may account for some of the phenotypes seen in FMRP deficient neurons particularly during development [32–35].

Despite the widespread gene expression of FMRP and the numerous potential mechanisms of action by which FMRP can impact the transcriptome, it was surprising that only very small changes in the mRNA levels of total or polyribosome associated mRNAs were seen in the *Fmr1*-KO mouse brain [36]. We and others have observed widespread, albeit very subtle, changes in mRNA levels in the absence of FMRP [22,23,37]. One possible cause behind the lack of detection of strong RNA changes is the cell type heterogeneity of brain tissue, where alterations in specific cell types could be masked in global measurements. Alternatively, the FMRP deficient brain could be inherently lacking strong changes at the transcriptome level, but instead display only mild RNA changes that can be challenging to detect using traditional

bulk tissue RNA-seq techniques due to low statistical power. Fortunately, both of these scenarios can be addressed using single-cell RNA-seq. Here we describe our effort to revisit this question using an unbiased approach to survey cell type specific transcriptomes in the *Fmr1*-KO mouse brain. We took advantage of the power of single cell RNA-Seq to determine which cells are affected by the lack of FMRP at an early postnatal development stage. The cell type specific alteration of the transcriptome is a sensitive reflection of the cellular status, and can serve as a first step towards an overview of the molecular impact of FXS.

We profiled the transcriptome of over 18,000 cells from the cerebral cortex of wild type (WT) and *Fmr1*-KO mice at postnatal day 5. We detected a heterogeneity in the response of different cell types to the loss of FMRP. In particular, we observed a stronger impact on the expression of mRNAs previously identified as FMRP binding targets in the brain [36], and we show that this effect is more prominent in neurons compared to other cells. We detect a divergent response of pathways downstream of mTOR signaling across different neuron subtypes, which suggests that excitatory neurons do not display a hyperactivation of this pathway. Taken together with the observed dysregulation of synaptic genes in astrocytes as well as neurons, our results suggest an impact in cell-cell communication that can result in a cortical environment of greater excitability.

## Results

### Cell type proportions are not impacted in the neonatal *Fmr1*-KO cortex

To capture early molecular events in the FMRP-deficient brain, we performed single cell RNA-Seq using the InDrop system [38], from the cortex of three *Fmr1*-knock out (*Fmr1*-KO) and three wild type (WT) FVB animals at postnatal day 5 (P5, Fig 1A, Methods), a critical period in cortical development for neuronal and synaptic maturation [39–41]. After stringent filtering, we obtained 18,393 cells for which we detected an average of 1,778 genes and 3,988 transcripts (see Methods). After unbiased clustering, we classified these cells into seven major cell types, with specific expression of established markers [42–45] (S1A Fig; Methods).

We did not detect a genotype bias in the clustering (S1B Fig). Additionally, all known major cell types were detected in both *Fmr1*-KO and WT animals, with consistent proportions across individual mice (S1C Fig). Consistent with previous reports [16], this indicates that *Fmr1*-KO mice do not have large scale cell differentiation deficits at this stage. In a second round of clustering, cells in each initial cluster were independently reclustered to better dissect cell subtypes. We identified a total of 18 distinct subpopulations (Figs 1B and S2A–S2D; Methods). These subpopulations included vascular cells (endothelial cells and pericytes), fibroblasts, ependymal cells, different neuron subtypes and glial cells (Fig 1B) [42–45]. The initial neuron cluster was subclassified into excitatory neurons (S2A Fig, Excitatory, $Nrgn^+$ $Lpl^+$ $Slc17a6^+$), immature interneurons (S2A Fig, Interneuron: $Gad1^+$ $Dlx1^+$ $Htr3a^+$), ganglionic eminence inhibitory precursors (S2A Fig, Ganglionic: $Ascl1^+$ $Dlx2^+$ $Top2a^+$), Cajal Retzius cells (S2A Fig, CR: $Reln^+$ $Calb2^+$ $Lhx5^+$) and subventricular zone migrating neurons (S2A Fig, SVZ: $Eomes^+$ $Sema3c^+$ $Neurod1^+$). We detected two immature astrocyte populations, including a small group with markers of reactive astrocytes (S2B Fig, Astrocytes_1: $Aqp4^+$ $Aldoc^+$ $Apoe^+$; Astrocytes_2: $Ptx3^+$ $Igfbp5^+$ $C4b^+$). Immune cells detected consist largely of microglia (S2C Fig, Microglia: $Cx3cr1^+$ $P2ry12^+$ $Csf1r^+$), but we also identified small clusters of T cells (S2C Fig, T Cells: $Cd3d^+$ $Ccr7^+$ $Cd28^+$), border macrophages (S2C Fig, Border Macrophages: $Lyz2^+$ $Msr1^+$ $Fcgr4^+$) and a population of microglia-like cells marked by high expression of Ms4a cluster genes (S2C Fig, Microglia Ms4a+: $Ms4a7^+$ $Ms46b^+$ $Mrc1^+$). The initial cluster containing oligodendrocytes were classified as mature oligodendrocytes (S2D Fig, Oligodendrocytes: $Mbp^+$ $Sirt2^+$ $Plp1^+$), oligodendrocyte progenitor cells (S2D Fig, OPC: $Pdgfra^+$ $Olig2^+$ $Sox10^+$) and a

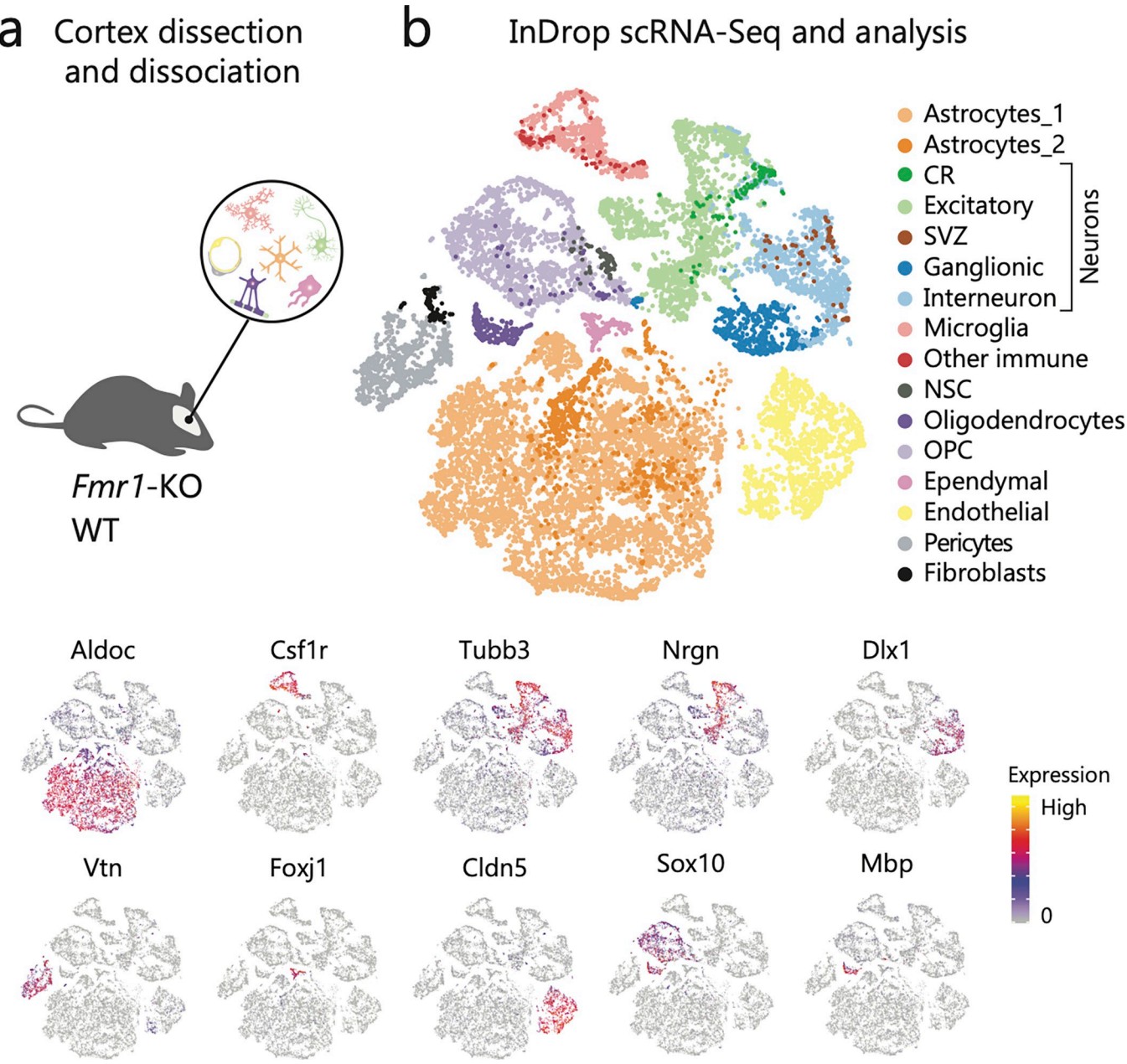

**Fig 1. Cortical cell types in *Fmr1*-KO and WT mice. a)** Overview of the sample collection **b)** tSNE representation of cells colored by cell type (top) and colored by expression level of example maker genes (bottom). tSNE = t-distributed stochastic neighbor embedding.

small group of neural stem cells (NSC: *Btg2*+ *Dll1*+ *Dbx2*+). We did not detect any cell number changes for the subtypes in the *Fmr1*-KO samples. Next, we investigated the gene expression landscape in the KO compared to WT.

## Loss of FMRP results in small expression changes across a broad and cell type specific spectrum of processes

We examined differential expression between *Fmr1*-KO and WT mice for each cell type identified. In total, we identified 1470 differentially expressed (DE) genes (FDR < 0.01 and absolute

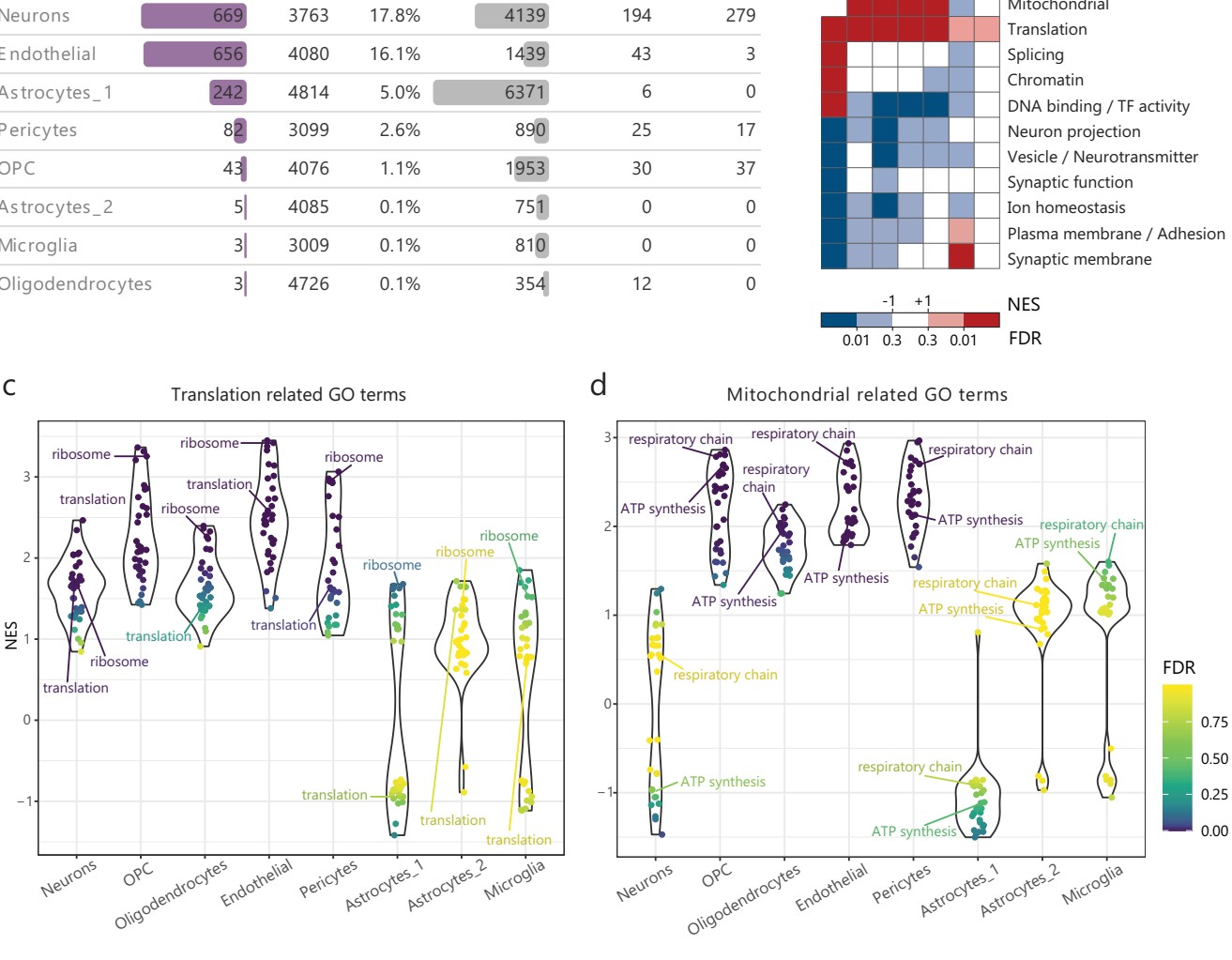

**Fig 2. Misregulated Gene Ontology categories per cell type reveal higher impact in *Fmr1*-KO neurons. a)** Differentially expressed (DE) genes identified per cell type, all results refer to expression in *Fmr1*-KO with respect to WT. Upregulated = Increased expression in *Fmr1*-KO. Downregulated = Decreased expression in *Fmr1*-KO. **b)** Summary diagram of enriched processes (dark red/blue; FDR<0.01) and trends (light red = NES>1, FDR<0.3; light blue = NES<-1, FDR<0.3; NES = normalized enrichment scores) in other cell types. **c-d)** Comparison of normalized enrichment scores (NES) across cell types for GO terms involved in translational (d: e.g. Ribosome, GO:0005840; Translation, GO:0006412) or mitochondrial (e: e.g. Respiratory chain, GO:0070469; ATP synthesis = ATP synthesis coupled electron transport, GO:0042773) processes.

fold change > = 1.15) in one or more cell types (Figs 2A, S3 and S1 Table). The majority of DE genes showed small fold changes (mean absolute fold change = 1.3x). The effect of FMRP loss seems drastically different across cell types with most differentially expressed genes being found in neurons and endothelial cells (Fig 2A). This result does not seem to be dependent on statistical power, as the number of DE genes detected did not correlate to the number of cells available or the number of transcripts detected for different cell types.

Given the broad but small effect size resulting from loss of FMRP, we focused on quantifying the impact on annotated pathways using gene set enrichment analysis (GSEA; see Methods). Most significantly enriched processes are dysregulated in *Fmr1*-KO neurons, further

suggesting that neurons are the cells most impacted by FMRP loss (Figs 2A and 2B, S4A–S4B and S2 Table). In all, we found that 473 categories were affected by FMRP loss. The vast majority (422) of these categories are affected exclusively in neurons, where most are downregulated (245 vs 177, S4B and S4D Fig) Fewer categories were upregulated, e.g. RNA splicing (Figs 2B, S4A and S4C), and genes in this category encode core spliceosomal proteins [46], including *Rbm39*, *Hnrnpm* and *Srsf3* (S5A Fig). Upregulation of these factors has been previously linked to an increased rate of proliferation [47–49].

Gene sets related to translation and mitochondrial processes (e.g. Translation, Ribosome proteins, Respiratory Chain Complex and Oxidative Phosphorylation) are the most upregulated in the *Fmr1*-KO brain. These pathways are consistently upregulated in many FMRP deficient cell types. Translational processes are significantly upregulated in neurons, vascular cells (endothelial and pericytes), oligodendrocytes and OPCs (Fig 2C). Mitochondrial pathways are upregulated in multiple cell types including OPCs, oligodendrocytes, endothelial cells and pericytes (Fig 2D). Interestingly, transcriptional activation of genes involved in ribosome biogenesis as well as oxidative phosphorylation are a hallmark of activation of mTOR signaling [50–52]. Thus, elevation of these processes could be reflective of the increase in mTOR signaling previously observed in FXS patients and *Fmr1*-KO mice, and upregulation of ribosome biogenesis could in turn contribute to increased translation in *Fmr1*-KO mice [53,54].

Processes related to synaptic signaling, vesicle transport and ion homeostasis tend to be downregulated across all brain cell types (Fig 2B; e.g. *Apoe*, *Sar1b*, *Clstn1*, *Rab5a* and *Sqstm1* involved in vesicle pathways, S4B Fig), suggesting that intracellular transport of molecules is impaired in multiple cell types. An interesting exception is the upregulation of synaptic signaling pathways in astrocytes (Fig 2B), which we discuss below.

Upregulation of ribosomal pathways and downregulation of synaptic pathways and cell surface protein encoding genes has also been observed in human ESC-derived FMRP-KO neurons and hiPSC derived FXS neurons [55]. In general, categories that are specifically downregulated in mouse *Fmr1*-KO neurons are also downregulated in human FXS and FMRP-KO cultured neurons (S6 Fig). These include functions related to synaptic and cell surface proteins, ion channels and neuron projections (S4D–S4E Fig). Overall, this indicates that the effect of FMRP loss is highly conserved between human and mouse neurons and potentially for other cell types as well.

Taken together, our results reveal a variable transcriptional landscape in the neonatal *Fmr1*-KO cortex, where not all cell types are affected equally. Factors involved in biomolecule production (protein and ATP synthesis) are upregulated in multiple cell types. Conversely, factors important for developing cellular communication (i.e. synaptic signaling and vesicle transport processes) and maintaining ion homeostasis are downregulated in multiple cell types, particularly in neurons (Fig 2B). The effects on these processes in multiple cell types in the *Fmr1*-KO cortex collectively point to enhanced growth and metabolism, and suggest defective network development through impaired cell-cell communication.

## FMRP bound mRNAs are at the core of the neuron response to FMRP loss

FMRP has been previously shown to directly bind hundreds of mRNAs in the mouse brain [12,36,56] or in cell lines [57,58]. The current model suggests that FMRP binding leads to translational repression of its target mRNAs [36,59], although this repression was only validated for a handful of direct targets [11]. In our scRNA-Seq data, FMRP bound mRNAs are enriched in all downregulated neuron specific processes (Fig 2B, $p < 10^{-27}$, hypergeometric). Strikingly, the top genes related to synapses and neuronal projections that are downregulated are also FMRP bound (S7A Fig): *Vamp2*, which encodes a key component of AMPA receptor

subunit vesicle trafficking [60]; *Camk2b* and *Camkk2*, which are both involved in the formation of dendritic spines [61]; and *Grm5*, which encodes the mGluR5 receptor, notably one of the main proposed therapeutic targets for FXS [62]. Likewise, most of the orthologous genes are also downregulated in human FXS neurons (S7B Fig). We therefore wondered whether FMRP bound transcripts are specifically affected, and are thus responsible for explaining the trends we observe.

We focused on the impact of FMRP loss on mRNA abundance of its direct binding targets across different cell types. Out of the 842 mRNAs bound by FMRP in the mouse brain [36], we detected 839 expressed by at least one cell type in our data. We compared the fold change between *Fmr1*-KO and WT for FMRP bound mRNAs to all other expressed mRNAs. Consistent with the strong effect of FMRP loss in neurons, FMRP bound mRNAs show a stronger downregulation in neurons compared to other expressed mRNAs (p < $10^{-57}$, Wilcoxon rank-sum), while for non-neuronal cell types there is little difference (Figs 3A and S8A).

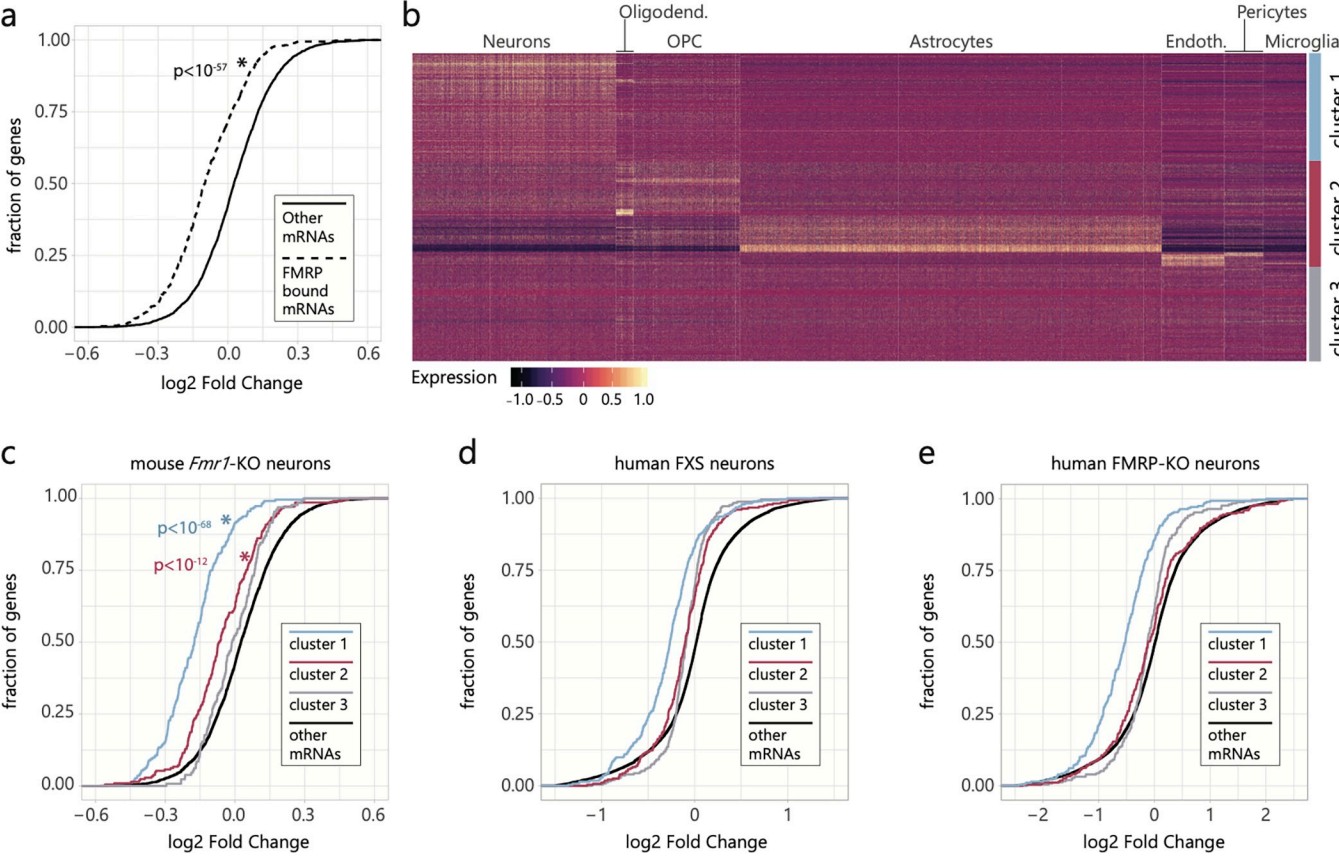

**Fig 3. FMRP bound mRNAs show greater downregulation in neurons. a)** Cumulative distribution plot of the log2 fold change between *Fmr1*-KO and WT gene expression in neurons. Dashed lines represent mRNAs annotated as FMRP bound by Darnell et al. expressed in that cell type. Full lines represent all other expressed genes in that cell type. **b)** Heatmap of expressed FMRP bound mRNAs in WT cortex cells. Genes were clustered based on their expression pattern across cell types and resulting clusters comprise genes expressed at higher levels in neurons (cluster 1), non-neuronal cell types (cluster 2) and without cell type specificity (cluster 3). **c)** Cumulative distribution plot of the log2 fold change between *Fmr1*-KO and WT gene expression in neurons. Colored lines represent each specific subset of expressed genes: Neuron FMRP bound mRNAs, which show highest expression in neurons (blue, cluster 1, Fig 3B); Non-neuronal FMRP bound mRNAs, which show highest expression in other cell types (pink, cluster 2, Fig 3B); Non-specific FMRP bound mRNAs (grey, cluster 3, Fig 3B), which are expressed at similar levels by all cortical cell types; and all other mRNAs expressed in neurons (black). **d-e)** Similar to (c) but displays the log2 fold change distribution for the homolog genes in each cluster of FMRP-bound mRNAs in human FXS neurons (d) or human FMRP-KO neurons (e). P-values refer to Wilcoxon rank-sum tests comparing the fold change distribution of each group of FMRP bound mRNAs to all other expressed mRNAs.

Despite the fact that FMRP bound mRNAs were identified in bulk tissue, they display cell type specific expression (Fig 3B), and therefore different subsets of FMRP targets could respond differently depending on the cell type. These mRNAs can be grouped in three sets (Fig 3B), the first set (35%) being most abundant and most affected in neurons (cluster 1, p < 10$^{-68}$, Wilcoxon rank-sum, Figs 3B–3C and S8B–S8C). Another set of 34% of FMRP bound mRNAs show a higher relative expression in non-neuronal cell types compared to neurons (Fig 3B), but are still highly expressed in neurons (S8B Fig) and their downregulation remains strongest in neurons (cluster 2, p < 10$^{-12}$, Wilcoxon rank-sum, Figs 3C and S8C). The remaining FMRP bound mRNAs are not expressed at high levels by any of the cell types analyzed, and do not seem to be affected by the absence of FMRP in neurons or other cells (cluster 3, Figs 3B–3C and S8B–S8C). Therefore, FMRP target mRNAs that are highly expressed in neurons, regardless of their expression levels in non-neuronal cells, are specifically downregulated in neurons. In cultured human FXS and FMRP-KO neurons, we observed the same trend of downregulation only for the cluster 1 gene orthologs (Figs 3D–3E and S9A–S9B), even though the average expression level of genes in the three clusters is similar in healthy human neurons (S9C Fig). We further examined if other published FMRP bound mRNA lists showed the same signal as seen for the Darnell *et al.* mRNAs, and reported the results for mouse and human neurons in S10 Fig and S4 Table. Three other lists show a significant enrichment for downregulated genes [12,56,63], which is not surprising given the large overlap between genes present in these lists and those reported by Darnell et al [36].

## Critical neural developmental pathways are nonuniformly dysregulated in excitatory versus inhibitory neurons

Many synaptic pathways are collectively impacted in *Fmr1*-KO neurons (S4D Fig). We examined if these and other pathways are uniformly or differently impacted in neuron subtypes. We focused on the two most abundant neuron subtypes: excitatory and interneurons (S1D Fig).

For one, the increased expression of genes related to ribosomes and translation (Fig 2C) is unique to interneurons. In fact, excitatory neurons show downregulation of these genes (Fig 4A). Secondly, mitochondrial pathways are also affected differently between the two subtypes. These pathways are downregulated in excitatory neurons while not changed in interneurons (Fig 4B). This reveals a critical difference in the effect of FMRP loss on different neuronal

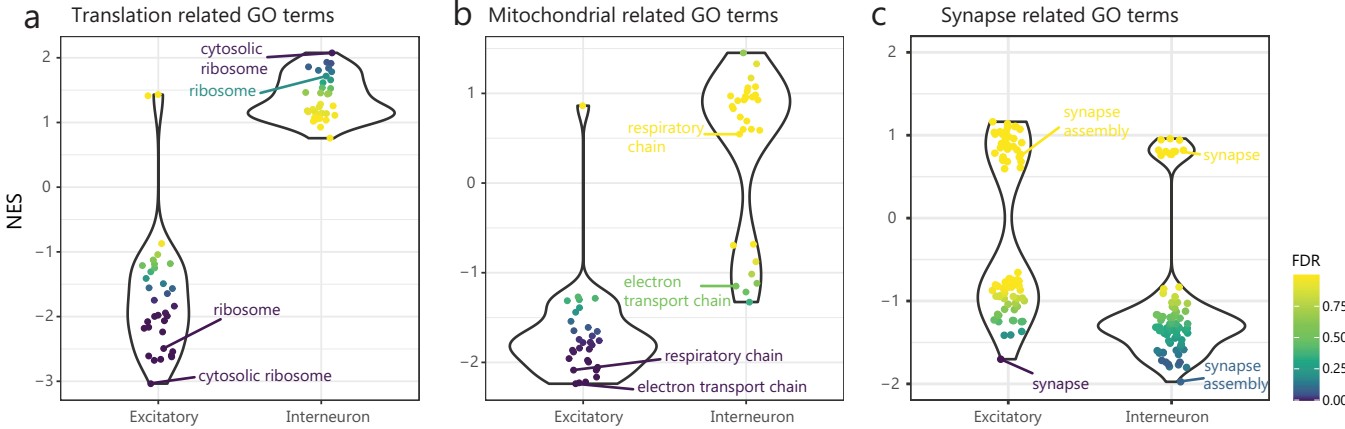

**Fig 4. Neuron subtypes show diverse misregulated genes.** Comparison of normalized enrichment scores (NES) across neuron subtypes for **a)** synaptic related GO terms (e.g. Synapse, GO:0045202; Synapse assembly, GO:0051963). **b)** translation related GO terms (c: e.g. Ribosome, GO:0005840; Cytosolic ribosome, GO:0022626) or **c)** mitochondrial function related GO terms (d: e.g. Respiratory chain, GO:0070469; Electron transport chain, GO:0022900).

subtypes. It has been observed that mTOR signaling is increased in *Fmr1*-KO mice [53,54] as well as in humans with FXS [64]. mTOR signaling results in increased metabolic activity and, in particular, in upregulation of translation activity (and ribosomal genesis) as well as in upregulation of mitochondrial processes [50,51]. We observe such an increase only in inhibitory neurons, suggesting that they are the subtype affected by the hyperactivation of mTOR signaling. On the other hand, the downregulation of these pathways in excitatory neurons suggests that in this subtype there may be a previously underappreciated dampening of mTOR activity. For neuronal projection and synaptic-related pathways downregulated in neurons as a group (Fig 2B), we observed a similar downregulation trend although for different individual GO terms in excitatory and interneurons (Fig 4C, e.g. Synapse in excitatory neurons and Synapse Assembly in interneurons), albeit the results are not significant in these subtypes likely due to the smaller number of cells examined (FDR<0.25, S2 Table).

Additionally, the top downregulated genes in interneurons are enriched for genes involved in pathways such as neuron maturation (GO:0042551, p<0.001, hypergeometric; e.g. *App*, *Nrxn1*; S11A Fig) and synaptic vesicle localization (GO:0097479, p<0.01, hypergeometric; e.g. *Adcy1* S11A Fig), which tend to be higher expressed at later stages of neuronal development [65–68]. Conversely, top downregulated genes in excitatory neurons (other than ribosomal and mitochondrial related genes), include many regulators of early stage neurogenesis (e.g. *Slc1a3*, *Tpi1* and *Lrrc17*; S11B Fig) [69–73]. The dysregulation of these gene groups suggests an impact on the development of both neuron subtypes at different stages.

Lastly, the most abundant neuron progenitor subtype, the ganglionic eminence inhibitory precursors, shows downregulation of synaptic pathways but upregulation of mitochondrial related pathways (S11C–S11E Fig). This effect in the *Fmr1*-KO ganglionic eminence inhibitory precursors indicate a possible divergence from typical development already at the neuronal precursor stage [74].

## Critical astrocyte-neuronal communications are disrupted by loss of FMRP

Many processes are dysregulated in both neurons and astrocytes, sometimes with opposite effects, revealing a surprisingly different impact of the loss of FMRP in these two cell types. Genes encoding for synaptic signaling (e.g. Integral component of synaptic membrane = GO:0099699; see S2 Table for others) and cell surface proteins (e.g. Intrinsic component of plasma membrane = GO:0031226; see S2 Table for others) are in general downregulated in neurons and other cell types, while upregulated in astrocytes (Fig 5A and S2 Table). The dysregulated genes in these processes include a large number of known receptor-ligand pairs involved in cross-cell signaling, and their dysregulation in multiple cell types could contribute to an imbalanced extracellular neurotransmitter and gliotransmitter homeostasis (Figs 3B–3C; leading edge analysis, GSEA).

Changes in astrocyte and neuronal communications likely result in an excitatory state: two ephrin receptor coding genes, *Ephb3* and *Epha4*, are upregulated in astrocytes (Fig 5B and S1 Table). These receptors are activated by ephrin-B (encoded by *Efnb3*) produced by neurons (Fig 5B, S1 Table), and this activation leads to enhanced release of D-serine (NMDA receptor coagonist) and glutamine (uptaken by excitatory terminals and converted to glutamate) [75,76]. Further, we also observe a downregulation of *Slc1a4* (*Asct1*; Fig 5B and S1 Table) in neurons, which could lead to reduced clearance of D-serine and an increase in extracellular excitatory neurotransmitters [77]. Additionally, we observed an upregulation of *Gabbr1* and *Gabbr2* in *Fmr1*-KO astrocytes, which encode two subunits of the GABA$_B$ receptors (Fig 5B and S1 Table). This could lead to enhanced glutamate release resulting from the activation of GABA$_B$ receptors [78].

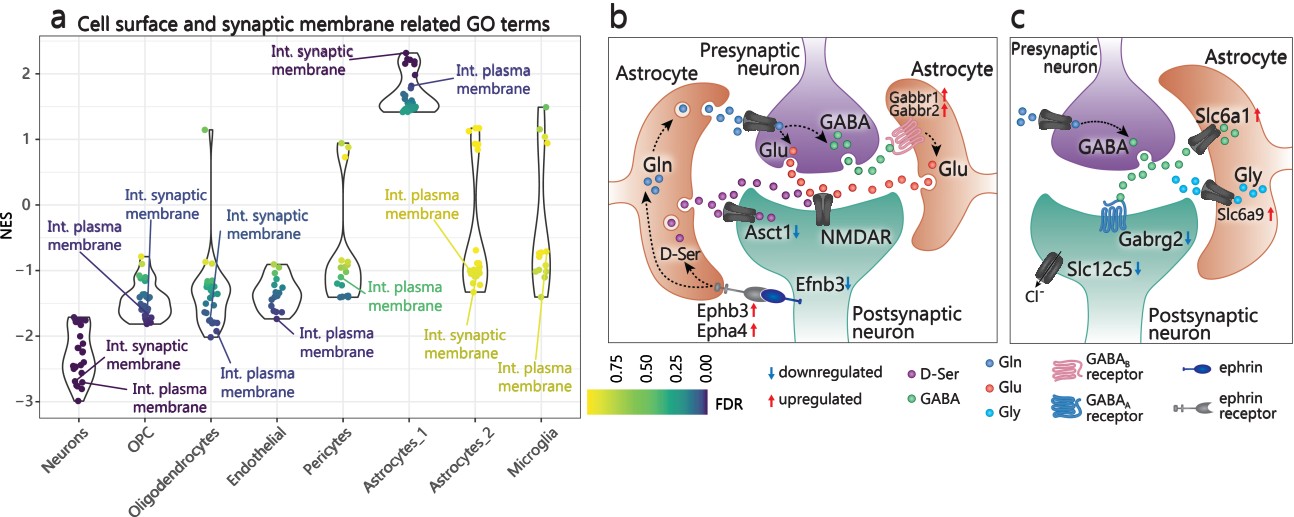

**Fig 5. Dysregulated genes in *Fmr1*-KO astrocytes and neurons involved in synaptic functions. a)** Comparison of normalized enrichment scores (NES) across cell types for all GO terms that show opposite signals in astrocytes and neurons (e.g. Int. synaptic membrane = Integral component of synaptic membrane, GO:0099699; Int. plasma membrane = Intrinsic component of plasma membrane, GO:0031226). **b-c)** Summary diagram of strongest dysregulated genes in astrocytes and neurons involved in synaptic signaling and regulation which suggest an increased release of excitatory (b) and a higher clearance of inhibitory (c) gliotransmitters and neurotransmitters in the synaptic environment. D-Ser = D-Serine; GABA = Gamma aminobutyric acid; Gln = Glutamine; Glu = Glutamate; Gly = Glycine.

On the other hand, our data also suggests a dampened inhibitory signaling resulting from dysregulated membrane and synaptic genes in both neurons and astrocytes. In astrocytes, genes such as such as *Slc6a1* (GAT-1), and *Slc6a9* (GlyT-1) are upregulated (Fig 5C and S1 Table), and this could lead to enhanced clearance of GABA and glycine (both inhibitory neurotransmitters) from the neuronal environment. In neurons, the downregulation of a GABA$_A$ receptor gamma subunit (*Gabrg2*) (Fig 5C and S1 Table) suggests their reduced sensitivity to environmental GABA inhibitory signals. Previously, in both fly and mouse models of FX, multiple subunits (including gamma) were reported to have reduced expression [79]. We additionally detected reduced expression of *Slc12c5* (KCC2) in *Fmr1*-KO neurons (Fig 5C and S1 Table), which encodes a transporter crucial for the switch of GABA from being excitatory to inhibitory [80]. This reduction could underlie the delayed switch in GABA polarity seen in *Fmr1*-KO mice [81].

Two of the most strongly downregulated genes in astrocytes are the two metallothioneins, *Mt1* and *Mt2* (S3 Fig and S1 Table), which are likewise critical to astrocyte-neuron interactions. Metallothioneins secreted by astrocytes play a key role in the protection of the central nervous system in response to injury, and neuronal recovery is impaired in their absence [82].

Collectively, these changes point to an imbalanced extra and intra-neuronal environment that favors excitation over inhibition (Figs 5B–5C). Indeed, cortical hyperexcitability is believed to be the biological basis of ASD and epilepsy, including FXS [83].

## Discussion

Our study presents the first attempt to dissect the cell type specific contributions to FXS pathology using the power of single cell RNA-Seq, and reveals cell type specific alterations that could be masked in global measurements. We found that FMRP loss results in small changes to individual gene expression overall, but is specific to core functional processes and is most severe in neurons. By dissecting cell type specific effects, we have described several novel

findings in FMRP deficient mice including: i) Higher impact of FMRP loss in neurons; ii) Potential differences in mTOR activity across different cell types; iii) Changes in abundance of cell-cell signaling genes that could increase environmental excitability. We further discuss these findings below.

## Impact of FMRP loss in neurons

Neurons are markedly the most impacted cell type, with synaptic and neuronal projection among the most strongly downregulated processes. Our data further points to multi-leveled transcriptome deficits (vesicle transport, cell morphology and receptors, affecting both pre and postsynaptic structures), which could explain known deficits of FXS neurites and synaptic development [4]. Neurons are also the only cell type where the set of mRNAs bound by FMRP in the mouse brain are distinctively downregulated, and we find no evidence that expression of FMRP protein partners contribute to this neuron specific effect (S12 Fig). The mechanisms that result in this cell type-specific and binding target-specific effect are likely miscellaneous, but some of the likely contributing mechanisms could be tied to reduced mRNA stability of FMRP targets [22,23,84,85], dysregulated mRNA export from the nucleus [86,87], dysregulated chromatin homeostasis which in turn can impact genome stability, RNA splicing, and transcription [25,88,89], or cascade effect downstream of dysregulated translational or signaling pathways, even if the direct effect may not be necessarily reflected at RNA level, e.g. the cAMP-cGMP pathway dysregulated in FXS [12,90] that can impact downstream transcription programs and neurodevelopment [91]. However, it remains unclear why FMRP loss affects FMRP targets most strongly in neurons. We hypothesize that this neuron specific downregulation of FMRP targets is due to the neuron specific expression of FMRP itself, and the loss of FMRP has a stronger impact on a subset of FMRP bound mRNAs. Although we did not detect a differential *Fmr1* mRNA expression across the major brain cell types (S12B Fig), FMRP protein was previously shown by multiple immunostaining studies to be predominantly expressed in neurons as opposed to other cell types in the brain [16,30,92,93]. Of note, the majority of the FMRP bound mRNAs show highly dynamic expression patterns across the embryo to postnatal development, in both neurons and in non-neuronal cells (S13A and S13B Fig). At postnatal day 5 (P5), the majority of these FMRP bound mRNAs start to be robustly expressed. Additionally, this subset of targets with high expression at P5 is enriched for genes related to synaptic functions (S13A Fig). This suggests that P5 is a critical developmental time point in FMRP mediated RNA homeostasis. It is possible that different FMRP bound mRNAs are downregulated at different developmental stages in mouse brains that lack FMRP, and that a different set of FMRP bound mRNAs may become of more relevance to the pathology at adult stages (S13A and S13B Fig). Finally, many of the neuronal downregulated pathways including neuron projection terms and vesicle transport were identified as downregulated both in RNA and protein levels by recent studies in human *in-vitro* neurons derived from embryonic or pluripotent stem cells [94,95]. This suggests that the downstream molecular consequences of the absence of FMRP in neurons are well conserved in mammals and that the wealth of altered pathways we observed in the *Fmr1*-KO mouse model can provide a valuable resource for future studies in FXS patients and the design of therapeutic approaches.

## Potential differences in mTOR activity

Multiple cell types display an upregulation of genes involved in translational and energy production processes, which are likely a downstream consequence of the increase in mTOR complex 1 signaling that has been previously detected in *Fmr1*-KO mice. The activation of this pathway is thought to be linked to the exaggerated mGluR5-LTD and a critical mechanism

behind FXS pathology [53,96]. We find that cortical excitatory neurons show a downregulation trend for both types of processes, indicating that they respond to the loss of FMRP differently than inhibitory neurons and possibly reflect a decrease in mTOR activity. Indeed, stimulation of the N-methyl-D-aspartate receptor (NMDAR) decreases mTOR signaling activity [97–99]. We find that NMDAR subunits are most highly expressed in excitatory neurons (S14A Fig) and one subunit, Grin2b, is upregulated in *Fmr1*-KO excitatory neurons (S14B Fig, FDR = 0.019). This increased expression of NMDAR in excitatory neurons, together with a gliotransmitter and neurotransmitter environment that favors activation of NMDAR (Fig 5B–5C), likely results in increased NMDAR signalling which suppresses mTOR activity and downstream pathways in excitatory neurons. On the other hand, inhibitory neurons exhibit increased metabolic gene expression which is concordant with mTOR hyperactivity. This implies that the current therapeutic strategies that target mTOR signaling for rescuing the cognitive and synaptic deficits in FXS may have drastically different effects in different neuron subtypes.

## Changes in abundance of cell-cell signaling genes and environmental excitability

A growing body of evidence supports the role of astrocytes in the pathology of neurological disorders [100,101]. Astrocyte involvement in FXS in particular was previously demonstrated by co-culture studies where neurons exhibited decreased levels of synaptic proteins and abnormal dendritic morphology when grown with astrocytes from *Fmr1*-KO mice [102]. Our analysis of the *Fmr1*-KO single cell transcriptome reveals that many key receptors and secreted proteins are misregulated in astrocytes. These alterations suggest that *Fmr1*-KO astrocytes contribute to an environment of increased excitability, and could also impact neuronal development. Dysregulation of many other genes that regulate cellular communication with its environment in diverse cell types may also contribute to the FXS phenotypes of mRNA translation and neurological defects. These include genes downregulated in oligodendrocytes, such as *Slc1a2* (GLT-1, S1 Table), involved in glutamate regulation and critical for white matter development [103,104]. *Slc7a5* (LAT-1, S1 Table) is downregulated in endothelial cells, and encodes a mediator of amino acid uptake that can impact mRNA translation in the brain, causing neurological abnormalities [105]. Astrocytes, in turn, show an increased expression of *Slc30a10* (ZnT10, S1 Table), which is responsible for $Zn^{2+}$ and $Mn^{2+}$ transport to the extracellular space [106]. High cellular levels of $Zn^{2+}$ and $Mn^{2+}$ induce the transcription of metallothionein mRNAs [107]. Castagnola *et al*. showed a global altered AMPA response in *Fmr1*-KO excitatory compared to inhibitory neurons [108]. AMPA receptor subunits need to be inserted to the postsynaptic membrane by vesicle trafficking factors such as Vamp2 to regulate the excitatory synapses [60]. We show that in *Fmr1*-KO neurons, the *Vamp2* mRNA was dysregulated (S7A Fig), which could lead to impaired trafficking of AMPA receptors, and in turn contribute to the excitatory-inhibitory imbalance in the FXS brain.

Our findings underscore the complex nature of the pathophysiology of FXS which involves the interaction of numerous cell types and multiple misregulated pathways. We hypothesize that it is this collective change that's driving the pathophysiology but not that of a few individual genes. Additionally, some of the findings described here have been independently observed in other human models. More recently, the development of human FXS forebrain organoids shows enhanced neuronal excitability by immunocytochemistry, and downregulated neurodevelopmental pathway genes, as well as upregulated protein translation and mitochondrial pathway genes by scRNA-Seq [109]. Together these studies provide invaluable resources for the FXS research community. We focused on the mouse cortex at P5, a critical period for

mouse cortical development, to identify changes that precede the onset of the FXS-like pheno-types. Future studies that profile earlier and later developmental stages would be invaluable to track the transcriptomic development of the disease, and would be instrumental for identifying developmental stage appropriate treatments. The mechanism by which cells, and neurons in particular are strongly affected by FXS is likely the result from a combination of cell intrinsic and extrinsic effects. To dissect these effects in the proper context, cell type specific KO models will be critical.

## Methods

### Ethics statement

All experimental procedures followed the animal care protocol approved by the University of Massachusetts Medical School Institutional Animal Care and Use Committee (IACUC).

### Mice

Wild type (WT; FVB.129P2-*Pde6b$^+$ Tyr$^{c-ch}$*/AntJ; JAX stock 004828) and *Fmr1* knockout (*Fmr1*-KO; FVB.129P2-*Pde6b$^+$ Tyr$^{c-ch}$ Fmr1$^{tm1Cgr}$*/J; JAX stock 004624) mice at postnatal day 5 (P5) of both sexes were used.

### Cortex dissection and dissociation

Three mice per genotype were used for single-cell collection. P5 mice were euthanized by decapitation and the brain was rapidly removed and placed in cold dissection buffer (1x HBSS). The brain was rapidly dissected to collect the cerebral cortex (both hemispheres) and the tissue was dissociated using the Papain dissociation system (Worthington Biochemical) following the manufacturer's protocol with a 30 minute incubation.

### InDrop collection and scRNASeq library preparation

Dissociated cortical cells were resuspended in PBS, filtered through a 70 micron cell strainer followed by a 40 micron tip strainer, and counted with a Countess instrument (Life Technologies). The cells were further diluted to the final concentration (80,000 cells/mL) with OptiPrep (Sigma) and PBS, and the final concentration of OptiPrep was 15% vol/vol. A total of 2,000 to 5,000 cortical cells were collected per mouse and processed following the InDrop protocol [38,110]. Final libraries containing ~1,200 cells were constructed and sequenced on an Illumina NextSeq 500 instrument, generating on average ~130 million reads per library (S3 Table).

### Data processing

**Cleanup, alignment and transcript quantification.** Sequencing reads were processed as previously described [111] and the pipeline is available through GitHub (github.com/garber-lab/inDrop_Processing). Briefly, fastq files were generated with bcl2fastq using parameters—*use-bases-mask y58n\*,y\*,I\*,y16n\*—mask-short-adapter-reads 0—minimum-trimmed-read-length 0—barcode-mismatches 1*. Valid reads were extracted using the barcode information in R1 files and were aligned to the mm10 genome using TopHat v2.0.14 with default parameters and the reference transcriptome RefSeq v69. Alignment files were filtered to contain only reads from cell barcodes with at least 1,000 aligned reads and were submitted through ESAT (github.com/garber-lab/ESAT) for gene-level quantification of unique molecule identifiers (UMIs) with parameters *-wLen 100 -wOlap 50 -wExt 1000 -sigTest .01 -multimap ignore -scPrep*. Finally, we identify and correct UMIs that are likely a result of sequencing errors, by

merging the UMIs observed only once that display hamming distance of 1 from a UMI detected by two or more aligned reads.

**Dimensionality reduction and clustering.** Gene expression matrices for all samples were loaded into R [112] (V3.6.0) and merged. Barcodes representing empty droplets were removed by filtering for a minimum of 700 UMIs observed. The functions used in our custom analysis pipeline are available through an R package (github.com/garber-lab/SignallingSingleCell). Using the raw expression matrix, genes were selected for dimensionality reduction with a minimum expression of 3 UMIs in at least 1% of all cells. From those, the top 30% of genes with the highest coefficient of variation were selected. Dimensionality reduction was performed in two steps, first with a principal component analysis (PCA) using the most variable genes and the R package *irlba* v2.3.3 [113], then using the first 7 PCs (>90% of the variance explained) as input to a t-distributed stochastic neighbor embedding (tSNE; package *Rtsne* v0.15) with parameters *perplexity = 30*, *check_duplicates = F*, *pca = F* [112]. Clusters were defined on the resulting tSNE 2-dimensional embedding, using the density peak algorithm [114] implemented in *densityClust* v0.3 and selecting the top cluster centers based on the γ value distribution ($\gamma = \varrho \times \delta$; $\varrho$ = local density; $\delta$ = distance from points of higher density). Using known cell type markers, clusters that corresponded to erythrocytes or potential cell doublets were excluded and the remaining cluster were used to define 5 broad populations (Immune, Vascular, Astrocytes, Oligodendrocytes and Neuronal). Erythrocyte marker genes (*Hba-a1*, *Hba-a2*, *Hbb-b1*, *Hbb-bh1*, *Hbb-bh2*, *Hbb-bs*, *Hbb-bt*, *Hbb-y*, *Hbq1a*, *Hbq1b*) were excluded from the resulting expression matrix. Raw UMI counts were normalized separately for each batch of libraries (S3 Table), using the function *computeSumFactors* from the package *scran* v1.12.1 [115], and parameter *min.mean* was set to select only the top 20% expressed genes to estimate size factors, using the 5 broad populations defined above as input to the parameter *clusters*. After normalization, only cells with size factors that differed from the mean by less than one order of magnitude were kept for further analysis ($0.1x(\Sigma(\theta/N)) > \theta > 10x(\Sigma(\theta/N))$; θ = cell size factor; N = number of cells). The normalized expression matrix was used in a second round of dimensionality reduction and clustering. The top 20% variable genes were selected as described above, and a mutual nearest neighbor approach was used to correct for batch effects implemented in the *fastMNN* function of package *batchelor* v1.0.1 [116]. To determine the number of dimensions used in the batch correction, the top 12 PCs that explained 90% of the variance were selected. The batch corrected embedding was used as input to tSNE, and the resulting 2D embedding was used to determine clusters as described above. Marker genes for each cluster were identified by a differential expression analysis between each cluster and all other cells, using *edgeR* [117], with size factors estimated by *scran* and including the batch information in the design model. Known cell type markers were used to determine 7 main populations (Immune, Endothelial, Pericytes, Ependymal, Astrocytes, Neuronal and Oligodendrocytes/OPCs). Each of these populations was independently re-clustered following the same procedure described above, to reveal more specific cell types and remove potential cell doublets (S1 Fig). All R scripts for the steps described here are available through GitHub (github.com/elisadonnard/FXSinDrop).

**Differential expression and GSEA analysis.** Differentially expressed (DE) genes between genotypes were identified per cell type using *edgeR*, with size factors estimated by *scran* and including the batch information in the design model. Only cell populations with at least 100 cells of each genotype were tested for differential expression, a threshold established empirically after downsampling both neurons and endothelial cells, which revealed a large increase in the variability of fold change values observed below this cell number. Only genes that had at least one UMI in 15% of the cells of that type were used in the analysis. Genes were considered

DE when they showed an FDR<0.01 and at least 1.15x fold change. Complete DE results can be found in S1 Table.

The gene set enrichment analysis (GSEA v3.0) [118] was performed using a ranked gene list constructed from the *edgeR* result. Genes were ranked by their reported fold change (logFC). GSEA was run via command line with parameters—*set_max 3000 -set_min 30*. The reference of GO term annotations for mouse Entrez GeneIDs was obtained from the *org.Mm.eg.db* package v3.8.2. GO terms were considered significantly altered if they showed an FDR<0.01. A total of 863 Gene Ontology (GO) terms had significant alteration (FDR<0.01) in one or more cell types. GO terms with high similarity were collapsed into 538 non-redundant categories (collapsedNAME, S2 Table), using the function *plot_go_heatmap* from the package *Signalling-SingleCell*, which compares the leading edge list of genes for each enriched pathway and creates a unified term if the overlap is greater than 80%. Collapsed categories were used to compare the results from different cell types (Figs 2A–2B and S3C and S3D Figs).

**Human ESC-derived neuron gene expression analysis.** All data for human cultured neurons was obtained from a recently published study [55]. Normalized counts and log2 fold changes are available as supplementary files in GEO accession GSE117248. Homolog genes were defined by the HomoloGene release 68 [119].

**FMRP bound mRNA expression analysis.** The normalized expression matrix was subset to select only WT cells and FMRP-bound mRNAs defined by Darnell *et al.* [36] expressed by at least one cell (n = 743). Cells were ordered first based on cell type assignment and hierarchically clustered based on the expression of these mRNAs (*dist* function *method = "euclidean"*; *hclust* function *method = "average"*). Genes were clustered using *k-means* (k = 7) followed by hierarchical clustering within the k-means cluster. Resulting clusters were merged based on the cell type pattern of expression to obtain the final three clusters (Neuronal, Non-neuronal, Non-specific). We examined other FMRP bound mRNA lists [12,56,63,120,121], described in [122], and their overlap with differentially expressed genes in this study are listed in S4 Table.

**Temporal expression analysis of FMRP targets.** The cell type specific expression analysis (Fig 3B) highlighted a group of FMRP bound mRNAs that are more abundantly expressed in P5 neurons. We assessed how this expression pattern changes across developmental stages, in order to identify critical FMRP targets at each stage. Using available single-cell RNA-Seq datasets, we performed a temporal analysis of the expression of FMRP bound mRNAs in the mouse cortex across four different developmental stages: E14 [42], P0 [42], P5 (this study, WT cells only) and P20 [44]. For each FMRP bound mRNA, we computed their aggregated expression in neurons or in non-neuronal cortical cells, and evaluated their relative expression in the same cell type across different time points. We focused on the three groups of genes identified in the P5 cell type analysis (clusters 1, 2 and 3; Fig 3B). Notably, very few FMRP bound mRNAs are expressed at higher levels in early development (E14.5 or P0), and these genes either show no particular functional enrichment with respect to the complete set of FMRP bound mRNAs, or are enriched for genes that encode nuclear proteins and DNA binding factors (S13 Fig), which could reflect the fact that the list of FMRP-bound mRNAs was defined using older mice (P14, P15 and P25; [36]). More significant neuron function enrichments, such as synaptic proteins, vesicle trafficking and ion channels are seen in the groups of genes that show highest expression at later postnatal timepoints (P5 and P20). Many of these genes appear to reach their maximal expression at P5, while others are expressed at even higher levels on the mature cortex, making this an interesting set of genes to explore in the absence of FMRP during later developmental stages.

## Supporting information

**S1 Fig. a)** tSNE representation of cells colored by major cell type. **b)** tSNE representation of cells separated and colored by genotype. **c)** Cell type composition of each collected sample.
(EPS)

**S2 Fig.** Unbiased sub-clustering results and subtype identification for each of the main cell types: **a)** Neurons; **b)** Astrocytes; **c)** Microglia and other immune cells; **d)** Oligodendrocytes and OPCs. tSNE = t-distributed stochastic neighbor embedding.
(EPS)

**S3 Fig. Plot of the log2 fold change (logFC) for each gene per normalized mean expression in aggregated WT cells (UMIs per million).**
(EPS)

**S4 Fig. a, b)** UpSetR diagrams showing overlap between GO terms identified as significantly upregulated (a) or downregulated (b)**. c)** Example GO terms upregulated specifically in mouse neurons: DNA binding (GO: 0003677; top), chromatin (GO:0000785; middle), and RNA splicing (GO:0008380, bottom). Center and right panels display the response for the same GO terms in human FXS and FMRP-KO cultured neurons, none of them showing significant change (FDR > 0.01). **d)** Example GO terms downregulated specifically in mouse neurons: synaptic membrane (GO:0097060; top), axon (GO:0030424; middle), and cation transport (GO:0098655; bottom). Center and right panels display the response for the same GO terms in human FXS and FMRP-KO cultured neurons, most of them showing significant change (FDR < 0.01). ES = GSEA Enrichment Score. NES = GSEA Normalized Enrichment Score.
(EPS)

**S5 Fig. a)** Expression in all mouse neurons for example genes upregulated in *Fmr1*-KO related to mRNA splicing. Numbers below boxplot correspond to numbers of cells present in each group (top) and median expression (bottom). **b)** Expression in all mouse single cells for example genes downregulated in multiple cell types related to vesicle pathways. Numbers below boxplot correspond to numbers of cells present in each group (top) and median expression (bottom).
(EPS)

**S6 Fig. a)** log2 fold change relative to healthy human cultured neurons for hiPSC derived FXS neurons and hESC derived FMRP-KO neurons reported in Utami *et al*. 2020. Genes depicted are homologs to the ones expressed in mouse neurons that show a significant change in one of the cultured human neurons (FDR < 0.01; n = 1953). Red colored genes show a significant upregulation in mouse *Fmr1*-KO (FDR < 0.01, log2FC > 0), while blue colored genes show a significant downregulation in mouse *Fmr1*-KO (FDR < 0.01, log2FC < 0). **b-e)** Comparison of normalized enrichment scores (NES) across human cultured neurons for GO terms involved in different processes which were significantly altered in mouse *Fmr1*-KO neurons. **b)** Translational (e.g. Ribosome, GO:0005840; Translation, GO:0006412). **c)** Mitochondrial (e.g. Respiratory chain complex I, GO:0045271; NADH dehydrogenase complex, GO:0030964). **d)** Synaptic functions (e.g. postsynaptic membrane, GO:0045211; Synaptic vesicle exocytosis, GO:0016079; Signal release from synapse, GO:0099643). **e)** GO terms which display opposite signals in mouse neurons and astrocytes (e.g. GABA-ergic synapse, GO:0098982; Synaptic membrane, GO:0097060).
(TIF)

**S7 Fig. a)** Expression in all mouse neuronal cells for example genes downregulated in *Fmr1*-KO involved in synaptic and neuron projection processes. Numbers below boxplot correspond to numbers of cells present in each group (top) and median expression (bottom). **b)** Expression of human orthologs for the same example genes in all bulk RNA-Seq replicates [55] of the human cultured neurons. Compared to healthy human ESC-derived neurons, the FXS derived neurons all display downregulation of the same genes, while human FMRP-KO derived neurons show upregulation of GRM5.
(EPS)

**S8 Fig. a)** Cumulative distribution plot of the log2 fold change between *Fmr1*-KO and WT gene expression in each cell type. Dashed lines represent genes annotated as FMRP bound by Darnell et al. expressed in that cell type. Full lines represent all other expressed genes in that cell type. **b)** Aggregated expression (UMIs per million) in WT cells for each cell type of all FMRP bound genes that are classified as neuronal (cluster 1), non-neuronal (cluster 2), or non-specific (cluster 3) in terms of their pattern of expression (heatmap clusters, Fig 3B). **c)** Cumulative distribution plot of the log2 fold change between *Fmr1*-KO and WT gene expression in each cell type. Colored lines represent each specific subset of expressed genes genes: Neuronal FMRP bound mRNAs, which show highest expression in neurons (blue, cluster 1, Fig 3B); Non-neuronal FMRP bound mRNAs, which show highest expression in non-neuronal cell types (pink, cluster 2, Fig 3B); Non-specific FMRP bound mRNAs (grey, cluster 3, Fig 3B), which are expressed at similar levels by all cortical cell types; and all other mRNAs expressed in neurons (black).
(EPS)

**S9 Fig. a-b)** Cumulative distribution plot of the log2 fold change in human FXS cultured neurons derived from iPSCs (a) or human FMRP-KO cultured neurons derived from hESCs (b). Dashed lines represent human homologs of mRNAs annotated as FMRP bound by Darnell et al., and full lines represent human homologs to the other expressed genes in mouse neurons. **c)** Expression (normalized counts) in healthy human cultured neurons derived from the H1 ESC cell line [55] for homologs of all FMRP bound genes that are classified as neuronal (cluster 1), non-neuronal (cluster 2), or non-specific (cluster 3) in terms of their pattern of expression (heatmap clusters, Fig 3B).
(EPS)

**S10 Fig. Cumulative distribution plot of the log2 fold change between FMRP deficient and WT gene expression in neurons, showing in colors different groups of genes annotated as FMRP bound by the respective publications cited. a)** Comparison in mouse *Fmr1*-KO neurons **b)** Comparison in human FXS cultured neurons derived from iPSCs [55] **c)** Comparison in human FMRP-KO cultured neurons derived from hESCs [55].
(EPS)

**S11 Fig. a)** Expression in all mouse interneurons for example genes downregulated in *Fmr1*-KO. Numbers below boxplot correspond to numbers of cells present in each group (top) and median expression (bottom). **b)** Expression in all mouse excitatory neurons for example genes downregulated in *Fmr1*-KO. Numbers below boxplot correspond to numbers of cells present in each group (top) and median expression (bottom). **c-e)** Comparison of normalized enrichment scored (NES) in ganglionic eminence progenitor neurons for **c)** synaptic related GO terms (e.g. regulation of trans-synaptic signaling, GO:0099177; regulation of postsynaptic membrane neurotransmitter receptor levels, GO:0099072). **d)** translation related GO terms (c: e.g. ribosome, GO:0005840; cytosolic ribosome, GO:0022626) or **e)** mitochondrial function related GO terms (d: e.g. inner mitochondrial membrane protein complex, GO:0098800;

respiratory chain, GO:0070469).
(EPS)

**S12 Fig. a)** Expression levels of known protein binding partners of FMRP across cell types. Aggregated bulk values were calculated per cell type (UMIs per million). **b)** *Fmr1* expression (UMIs) in WT cells of each cell type.
(EPS)

**S13 Fig. a-b)** Clustered expression of FMRP-bound mRNAs through developmental stages in neuronal cells (a) or all non-neuronal cortical cells (b). Clusters on the left are the same identified in Fig 3B, and refer to the expression pattern at postnatal day 5 (P5). Labels on the right of heatmaps represent significantly enriched GO terms for genes in each subcluster, with respect to all FMRP-bound mRNAs (Darnell *et al.*).
(EPS)

**S14 Fig. a)** tSNE representation of neurons colored by expression level of NMDAR subunits detected, Grin1 (left) and Grin2b (right), cells are separated by their genotype indicated above. **b)** Expression level (normalized UMIs) in *Fmr1*-KO or WT excitatory neurons for Grin2b. Numbers below boxplot correspond to numbers of cells present in each group (top) and median expression (bottom).
(EPS)

**S1 Table. Unfiltered differential expression analysis results for all genes tested in each major cell type.** The final columns indicate for each gene if it was found to bind FMRP by the study indicated in the column name (Y = FMRP binding target). logCPM = edgeR reported average log2-counts-per-million; LR = edgeR reported likelihood ratio statistics; PValue = the two-sided p-value reported by edgeR; FDR = adjusted p-value reported by edgeR.
(TSV)

**S2 Table. Unfiltered GSEA results showing all GO categories tested in each cell type.** NAME = GO identifier for the category tested; SIZE = Number of genes in the gene set after filtering out those genes not in the expression dataset; ES = Enrichment score for the gene set; that is, the degree to which this gene set is overrepresented at the top or bottom of the ranked list of genes in the expression dataset; NES = Normalized enrichment score; that is, the enrichment score for the gene set after it has been normalized across analyzed gene sets; NOM p-value = Nominal p value; that is, the statistical significance of the enrichment score; FDR q-value = False discovery rate; that is, the estimated probability that the normalized enrichment score represents a false positive finding. FDR q-value = False discovery rate; RANK AT MAX = The position in the ranked list at which the maximum enrichment score occurred; LEADING EDGE = Displays the three statistics used to define the leading edge subset (Tags: The percentage of gene hits before (for positive ES) or after (for negative ES) the peak in the running enrichment score; List: The percentage of genes in the ranked gene list before (for positive ES) or after (for negative ES) the peak in the running enrichment score; Signal: The enrichment signal strength that combines the two previous statistics); collapsedNAME = non-redundant category name for GO terms with high overlap in genes (see Methods); signal = text indicating the direction of the gene expression change (upreg = genes tend to be overexpressed in FMRP-KO; downreg = genes tend to be downregulated in FMRP-KO); Description = Full GO category name; core_enrichment = list of Gene IDs for the genes present in the leading edge that are annotated in the GO category tested.
(TSV)

**S3 Table. Metadata for each mouse cortex single cell RNA-Seq library generated.**
(TSV)

**S4 Table. Overlap results between genes detected as downregulated or upregulated in one or more cell types and previously published lists of mRNAs bound by FMRP.** Significant overlaps are shaded in green.
(PDF)

## Acknowledgments

We thank Paul Greer and members of the Garber lab for critical reading of the manuscript and thoughtful discussions. We thank Paulo Ortega for assistance with figures.

## Author Contributions

**Conceptualization:** Elisa Donnard, Huan Shu, Manuel Garber.

**Data curation:** Elisa Donnard, Huan Shu.

**Formal analysis:** Elisa Donnard, Huan Shu.

**Funding acquisition:** Manuel Garber.

**Investigation:** Elisa Donnard, Huan Shu.

**Methodology:** Elisa Donnard.

**Project administration:** Manuel Garber.

**Resources:** Manuel Garber.

**Supervision:** Manuel Garber.

**Validation:** Elisa Donnard, Huan Shu.

**Writing – original draft:** Elisa Donnard, Huan Shu, Manuel Garber.

**Writing – review & editing:** Elisa Donnard, Huan Shu, Manuel Garber.

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
