## [Decision Letter · Decision Letter 0]

8 Sep 2021

Dear Dr Donnard,

Thank you very much for submitting your Research Article entitled 'Single Cell Transcriptomics Reveals Dysregulated Cellular and Molecular Networks in a Fragile X Syndrome Model' to PLOS Genetics.

The manuscript was fully evaluated at the editorial level and by independent peer reviewers. The reviewers appreciated the attention to an important problem, but raised some substantial concerns about the current manuscript. Based on the reviews, we will not be able to accept this version of the manuscript, but we would be willing to review a much-revised version. We cannot, of course, promise publication at that time.

If you decide to revise the manuscript for further consideration at PLOS Genetics, please aim to resubmit within the next 60 days, unless it will take extra time to address the concerns of the reviewers, in which case we would appreciate an expected resubmission date by email to plosgenetics@plos.org.

[LINK]

We are sorry that we cannot be more positive about your manuscript at this stage. Please do not hesitate to contact us if you have any concerns or questions.

Yours sincerely,

Mingyao Li

Associate Editor

PLOS Genetics

Gregory Barsh

Editor-in-Chief

PLOS Genetics

Reviewer's Responses to Questions

**Comments to the Authors:**

Reviewer #1: “Single Cell Transcriptomics Reveals Dysregulated Cellular and Molecular Networks in a Fragile X Syndrome Model” is an interesting manuscript. However, it seems that the authors do not have a clear overview of the Fragile X literature. This – in my opinion – negatively impacts not only the introduction but the analysis of the results and the discussion. My main suggestion is to reconsider all aspects of this manuscript in the light of a larger literature analysis.

For instance:

FMRP is not only a repressor but also an activator of translation: Bechara et al., PLoS Biology 2009; Ethan J. Greenblatt, Allan C. Spradling, Science 2018; Maurin & Bardoni , Front Mol Bioscience 2018.

One pathway that is critically modified in Fragile X is the cAMP-cGMP, as shown by multiple studies: (see a recent selection) Gurney et al., Sci Rep, 2017; Maurin et al., NAR 2018; Maurin et al., Cerebral Cortex 2019; Gurney et al., J Med Chem. 2019; Delhaye & Bardoni, Mol Psychiatry, 2021; Berry-Kravis EM et al., Nat Med, 2021

If the authors have results supporting these findings they have to underline them, if they do not have them, it is a critical point to be discussed.

The implication of FMRP in excitatory/inhibitory balance has been already considered in another study using single-cell RNAseq: Castagnola et al., Genome Res., 2020. These data should be mentioned by the authors who, in addition, should compare they results with those of Castagnola et al and discuss them.

Several previous analyses have defined by CLIP the target mRNAs of FMRP (please see for an exhaustive list in Richter JD & Zhao X Nat Rev Neurosci. 2021). It would be important to compare mRNAs whose expression is found altered here by Donnard et al., with the various lists found by different authors in order to understand :

i) The possible mechanism of action of FMRP;

ii) The alterations that are directly caused by the absence of FMRP and those that are due to a “cascade” effect.

-The mechanism of action of FMRP is quite complex, that seems here oversimplified

In the discussion the authors should underline the novelty of their study not only as a technological approach but at the level of study of the pathways involved in the pathophysiology of Fragile X. For instance, Figure 5 illustrates findings already known 5-6 years ago.

Minor: please correct polysome with polyribosomes

Reviewer #2: In this manuscript, the authors used single cell RNA sequencing to profile an FMRP deficient mouse model for Fragile X syndrome. Their findings suggest that FMRP loss affect cell-cell communication, thus resulting in a cortical environment of greater excitability. The study provides an useful resource for investigating molecular basis of Fragile X syndrome.

While the data are interesting, they are solely transcriptomics (mRNA) based. For many of the dysregulated genes, such like Ephb3 and Epha4 (upregulated in astrocytes), downregulation of Slc1a4 in neurons, upregulation of Gabbr1 and Gabbr2 in Fmr1-KO astrocytes, Slc6a1 (GAT-1), and Slc6a9 (GlyT-1) upregulated in astrocytes, downregulation of a GABAA receptor gamma subunit (Gabrg2) and reduced expression of Slc12c5 (KCC2) in Fmr1-KO neurons, the authors will need to use additional approach (immunocytochemistry, Western blot or others) to validate or assess the gene expression changes at protein levels. This is to further confirm that the dysregulated gene expression is able to reach the protein level to function biologically.

Reviewer #3: 

Donnard et al performed single cell RNA sequencing (scRNA-seq) with postnatal day 5 Fmr1 KO cerebral cortex to understand the mechanisms by which Fmr1 regulates FXS. The analyses generated some useful information, but this reviewer finds that the manuscript needs to provide rigorous statistical improvement and more experimental results.

1. To get more comprehensive results, the sc-RNA seq should be also done with brains at more mature stage at least 4-8 weeks old. Postnatal day 5 is estimated as gestational week 21 (Clancy, 2007). The usual diagnosis time and the average age of first concern is 12 months based on the CDC report. The authors could show if the DE patterns are conserved or changed. If changed, what would be the functional relationship between the DE and the phenotypes? 

2. ìWe examined differential expression between Fmr1-KO and WT mice for each cell type identified. In total, we identified 1470 differentially expressed (DE) genes (FDR < 0.01 and fold change >=1.15) in one or more cell types. The majority of DE genes showed small fold changes (mean fold change = 1.3x).î

This reviewer is most concerned about getting conclusion with the DE with fold change >=1.15. With such a small fold change, many of them just could be from biological sample or experimental batch variations. 

3. Given the nature of the data with such a small DE FC, the authors should show separate tSNE plots for Wt versus KO in all tSNE related figures. It wonders whether there is no distinct cell cluster(s) in fmr1 KO compared to control. Even with overall similar presentation in tSNE plot, Wt and KO could show some differential proportion change better if they are presented separately. 

4. Throughout the entire manuscript, the authors should provide the adjusted p values or FDR values of the DE genes. The authors provided p-vlaue for enrichment analysis, but did not mention statistical information for the analysis with differential expression genes. Without the proper significance, the claim of the authors could not be reasonable. The authors adopted FDR<0.25 0r 0.3 in many places.

5. Most of the violin plots in Figure S2d,e and S4c, do not look differential and significant to me. The results of bulk RNA-seq data seem to be driven by one very high outlier. The current violin plot is distracting since the outlier drags the boxes very high. I would suggest to use a boxplot to re-draw the results.

6. In Figure 4a, if we look at the terms with FDR<0.05, excitatory and interneuron do not show much difference.

7. In many places, the authors did not present Wt versus KO data side by side, which would not tell better the differential effect by loss of FMRP. For example, the authors should show the developmental change in the expression of specific subset of genes in both Wt and KO in Figure S6d, e and f or DE along with significant p-value. The authors should already have these data and could utilize the data fully to claim better. I believe people in FXS research would be more interested in the changes happening in KO. 

8. ìThe cell type specific alteration of the transcriptome is a sensitive reflection of the cellular status, and can serve as a first step towards an overview of the molecular impact of FXS.î If this is the purpose of this manuscript, cell-type specific KD of Fmr1 including neuron, endothelial cell and astrocytes should be shown. It is likely that the observed DE patterns of this manuscript is combined effects of cell intrinsic and extrinsic functions of Fmr1 and/or combined effects of multiple cells lacking Fmr1. Some experimental verification that can support the authorsí conclusions should be presented. Otherwise, enthusiasm on this manuscript would be minimal.

9. The arrangement of Supplemental figures is not easy to follow. The orders of Supplemental figures should go in parallel with the main text.

10. Some of the figures and Supplemental figures were not cited in the main text, for example, Figure S4a.

11. There is no Figure 2f in Figure 2, which is cited in the main text 

12. There is no reference (Utami et al, 2020 )cited in the Suppl figure legend. If the data was from Utami, 2019, the cells must be hESC, not iPSC derived neuron.

13. The comparison between mouse brain and human should be with human post mortem or at least forebrain organoids at comparable developmental stage.

**Have all data underlying the figures and results presented in the manuscript been provided?**

Reviewer #1: Yes

Reviewer #2: Yes

Reviewer #3: None

PLOS authors have the option to publish the peer review history of their article (what does this mean?). If published, this will include your full peer review and any attached files.

Reviewer #1: No

Reviewer #2: **Yes: **Hansen Wang

Reviewer #3: No

---

## [Decision Letter · Decision Letter 1]

23 Dec 2021

Dear Dr Donnard,

Thank you very much for submitting your Research Article entitled 'Single Cell Transcriptomics Reveals Dysregulated Cellular and Molecular Networks in a Fragile X Syndrome Model' to PLOS Genetics.

The manuscript was fully evaluated at the editorial level and by independent peer reviewers. The reviewers appreciated the attention to an important problem, but raised some substantial concerns about the current manuscript. Based on the reviews, we will not be able to accept this version of the manuscript, but we would be willing to review a much-revised version. We cannot, of course, promise publication at that time.

If you decide to revise the manuscript for further consideration at PLOS Genetics, please aim to resubmit within the next 60 days, unless it will take extra time to address the concerns of the reviewers, in which case we would appreciate an expected resubmission date by email to plosgenetics@plos.org.

[LINK]

We are sorry that we cannot be more positive about your manuscript at this stage. Please do not hesitate to contact us if you have any concerns or questions.

Yours sincerely,

Mingyao Li

Associate Editor

PLOS Genetics

Gregory Barsh

Editor-in-Chief

PLOS Genetics

Reviewer's Responses to Questions

**Comments to the Authors:**

Reviewer #1: In this manuscript the authors profiled by single cell RNA-seq WT and Fmr1-KO mouse brain cortex at postnatal day (PND) 5. They found a very high heterogeneous expression of target mRNAs of FMRP depending on the cell type. Very likely, the authors analyzed by single cell RNA-seq mouse brains at PND 5 for technical reasons (neuronal dissociation is easy at that age). Since it is known that the highest expression of FMRP is between 7 and 14 PND, I would suggest to tone down the discussion about the importance of post-natal-day 5 in FXS (pag. 12).

Furthermore, they compared their RNA dataset with FMRP RNA targets obtained in 1-month old mouse brain (Darnell et al., 2011). I consider this as an inconsistency since other studies identified FMRP RNA targets (or proteins whose expression is linked to the presence of FMRP, e.g. Tang et al., Proc Natl Acad Sci U S A. 2015 112(34):E4697-706) during the earliest phases of development. For this reason, at the first round of revision I asked to consider other datasets of FMRP targets in addition to the one they used (Darnell et al., Cell 2011). This was only superficially done in the revised manuscript by adding the incomplete table S4. This table is incomplete since, for instance, two important studies such as Sawicka et al, Elife (2019)n8:e46919 and Miiyashiro et al., Neuron (2003) 37(3):417-31 are missing. Please add those datasets to your Table S4.

The same in figure S8 a-b, please repeat the analysis shown with datasets other than the one published in 2011 by Darnell.

Description of Supplementary Tables is just minimal, a lot of abbreviations are not explained in the legend. For instance, in Table S1 what does clip mean? Cross-linking ImmunoPrecipitation? In this case, please indicate results of each clip as in Table S4 and in Figure S8.

The references’ list is totally confused and imprecise.

Many papers mentioned in the text or in the supplementary material do not appear in the final list.

3 examples:

Li, et al., (2020) Genome Research 30 (3): 361–74.

Castagnola et al. (2020) Genome Research 30(11):1633-1642.

Ritcher & Zhao (2021) Nat Rev Neurosci, 22(4):209-222

While other articles present in the reference list are not mentioned in the text.

2 examples:

Gurney et al., (2017) Scientific Report, 7 (1) 14653

Berry Kravis et al., (2021) Nature Medicine, 27 (5): 862–70.

Furthermore, at pag 11 (7th line from the bottom) two references: Berry-Kravis 1992 and Maurin 2018 are wrongly cited. The right references (considering the sentence written by the authors) should be Berry Kravis et al., Nature Medicine 2021 and Gurney et al, Scientific Report, 2017.

Minor point:

Several typos are present, e.g. Pag 13 …individual genes. And some…

I do not understand the sentence: Pag. 12: The neuron specific downregulation of FMRP targets upon loss of FMRP is therefore potentially the result of higher levels of FMRP protein expression.

Reviewer #2: In general, I am satisfied with the revision.

Reviewer #3:

Donnard et al performed single cell RNA sequencing (scRNA-seq) with postnatal day 5 Fmr1

KO cerebral cortex to understand the mechanisms by which Fmr1 regulates FXS. The

analyses generated some useful information, but this reviewer finds that the manuscript

needs to provide rigorous statistical improvement and more experimental results.

1. To get more comprehensive results, the sc-RNA seq should be also done with brains at

more mature stage at least 4-8 weeks old. Postnatal day 5 is estimated as gestational week

21 (Clancy, 2007). The usual diagnosis time and the average age of first concern is 12

months based on the CDC report. The authors could show if the DE patterns are conserved

or changed. If changed, what would be the functional relationship between the DE and the

phenotypes?

We appreciate the reviewer's point. While it is true that the disease in humans is clinically

diagnosed at a later age compared to the one we have examined, our study focused on

identifying molecular phenotypes that are present even if sub-clinically at earlier

developmental stages. Our goal w as to identify molecular changes that precede disease

onset and that therefore may eventually help to provide novel therapeutic clues. To this end,

we focused on postnatal day 5, which is a known “critical window” in mouse cortical

development, as evidenced by many other studies (Farhy-Tselnicker and Allen 2018; Harlow

et al. 2010; Nomura et al. 2017), and which has for that same reason been the focus of FXS

related studies on which we based ours (Darnell et al. 2011). While a study that looks at later

stages of brain development would also be fascinating, our goal was to focus on the effects

of FMRP loss on the developing brain with the goal to open new avenues of investigation and

generate a resource that new studies can build upon. We have better explained our

motivation for looking at the developing brain in Discussion.

>> The authors explained their goal more clearly.

2. “We examined differential expression between Fmr1-KO and WT mice for each cell type

identified. In total, we identified 1470 differentially expressed (DE) genes (FDR < 0.01 and

fold change >=1.15) in one or more cell types. The majority of DE genes showed small fold

changes (mean fold change = 1.3x).”

This reviewer is most concerned about getting conclusion with the DE with fold change

>=1.15. With such a small fold change, many of them just could be from biological sample or

experimental batch variations.

The fascinating observation of ours and other studies is that loss of FMRP does not cause a

large effect on the mRNA levels of most genes, however, the effect is consistent across most

genes within affected pathways. We did not try to make claims about any particular gene and

in fact, we agree with the reviewer in that conclusions about any particular gene may be

challenging at this level, and only applied this low threshold to define the numbers in Figure

2a. Our analysis instead focused either on specific genes with larger fold changes or on

pathways or gene sets with similar function where the overall trend is highly significant. The

overall small effect size of FMRP loss was one of the motivations of carrying out a single cell

analysis of FXS loss of function. In this version we have made an effort to better make this

point in the introduction: that the small effect size of the transcriptional changes detected is

expected, given that more robust changes would have been observed also by previous

studies which relied on bulk analysis of gene expression.

>> I understand that there could be masking effect in overall change by FMRP loss in bulk RNAseq.

However, with shingle cell RNAseq, in specific cell types, it is more probable to have larger effect, that is why single cell analysis is needed.

We want to stress that our approach to look at functionally related genes using gene set

enrichment analysis (GSEA) does not rely on a cutoff for selecting differentially expressed

genes, but instead examines the fold changed ranked gene list to identify any coordinated

expression change involving genes that belong to the same pathways or categories. As a

result, the genes in the pathways identified show a consistent (and significant compared to

other lists of genes) up or down regulation, which is a strong indicator of a biologically

relevant change resulting from the loss of FMRP.

>> I understand the author’s claim, but I still do not buy it fully. With the sets with statistically insignificant genes even with high fold change rank, the further analysis of GSEA would not be meaningful. Throughout the entire manuscript, it should be clearly stated that the criteria for the selected gene sets used for GSEA.

3. Given the nature of the data with such a small DE FC, the authors should show separate

tSNE plots for Wt versus KO in all tSNE related figures. It wonders whether there is no

distinct cell cluster(s) in fmr1 KO compared to control. Even with overall similar presentation

in tSNE plot, Wt and KO could show some differential proportion change better if they are

presented separately.

The reviewer raises a valuable question, we did in fact provide a figure of the tSNE

representation of the cells colored by their genotype in Figure S1b. As we discussed in the

text, there were no observable differences or clusters that were formed by cells of only one

genotype. This is not encouraging, considering the analysis relies on a set of genes that

exhibit the highest variability across cells, and this is largely dominated by the cell type

specific signal. Likewise, we examined these genotype plots for every subclustering analysis

performed, using each cell type independently, and found no consistent differences that

signalled a genotype difference in proportions. Again, this is not surprising given that the

changes we observed in expression are moderate. We provided both the code to generate

these plots (https://github.com/elisadonnard/FXSinDrop) as well as the single cell matrix

used as input (GSE147191).

>> It was asked to present separate tSNE plots of wt versus fmr1 KO, not overlapped in different colors to show clear gene expression change in each cell types in single cell analysis.

4. Throughout the entire manuscript, the authors should provide the adjusted p values or

FDR values of the DE genes. The authors provided p-vlaue for enrichment analysis, but did

not mention statistical information for the analysis with differential expression genes. Without

the proper significance, the claim of the authors could not be reasonable. The authors

adopted FDR<0.25 0r 0.3 in many places.

Unless specified, we used an FDR of 0.01 or less in our comparisons. However, as the

reviewer points out, we did in some cases want to point out specific examples or trends that

were significant in one cell type but not others, yet the trends were similar. Specifically:

● In figures describing GSEA results we report all categories that reach our FDR level

in at least one cell type, but also report their enrichment and FDR in all other cell

types (e.g. Figure 2b legend, where we specifically say the FDR<0.3 colors refer to

trends in other cell types). This is the same reason why categories reported in Figures

2c-d, 4, 5a, S5b-4 and S9c-e) may not reach the 0.01 cut-off for some cell types.

Every category has reached this significance in at least one cell type. We hope that

the reviewer agrees with us that it is informative to report how every category that is

disrupted in one cell type is affected on the other cell types.

● When we show the GSEA results for synapse related pathways (which are

significantly downregulated in all neurons) in excitatory and inhibitory neuron

subtypes. This panel is intended to show a similar trend between the two neuronal

subtypes, without claiming the significance of the change. We have made this clearer

in the text as well.

● When we discuss potential mechanisms for disruption of mTOR activity (Discussion)

we mention that Grin2b is upregulated in excitatory neurons (FDR = 0.019). We

explicitly point out that this gene is upregulated (revised Figure S12b) but it just barely

misses our significance cut-off.

5. Most of the violin plots in Figure S2d,e and S4c, do not look differential and significant to

me. The results of bulk RNA-seq data seem to be driven by one very high outlier. The current

violin plot is distracting since the outlier drags the boxes very high. I would suggest to use a

boxplot to re-draw the results.

We thank the reviewer for raising the concern that for the individual genes shown in Figure

S4, violin and point plots are difficult to read, and therefore we have changed to boxplots as

recommended. These genes are indeed significantly different between the WT and FK,

although not all of them show a large degree of change - the dominant majority of the DE

genes as we have discussed show moderate fold changes. Nevertheless, we chose to

display the comparison of these individual genes as a few examples from the more relevant

Gene Ontology categories, so that the readers could associate these pathways to key genes

of interest.

With respect to the human bulk RNA-Seq data (now Figure S6b), which are sourced from

(Utami et al. 2020) (the citation is included in the legend for better clarification), we agree that

one WT sample seems to be an outlier and shows much higher expression than other WT

samples for the genes shown. However, after removing this outlier, there is still a

considerable reduction in expression of these genes in the FXS neurons (average log2 fold

change -0.7), and more pronounced in the FMRP-KO derived neurons.

>> The new boxplot presents better.

6. In Figure 4a, if we look at the terms with FDR<0.05, excitatory and interneuron do not

show much difference.

We thank the reviewer for making an insightful observation. It is indeed our intention to show

that for the synaptic related GO terms (Figure 4a), excitatory and inhibitory neurons behave

very similarly. This is intended to contrast with the divergent behavior of the excitatory and

inhibitory neurons for translation and mitochondria related GO terms (Figure 4b-c). We intend

to show GO terms that behave similarly and differently to present this overview that

excitatory and inhibitory neurons share some common responses but also have some

divergent responses. However we have moved Figure 4a to 4c, and modified the text of this

section to further clarify the intention.

>> However, it seems that the excitatory and interneurons seem to have a bit specific trend. For example, significant excitatory GO terms are more in synapse and significant interneuron GO terms are more specifically in synapse assembly, not just synapse. 

7. In many places, the authors did not present Wt versus KO data side by side, which would

not tell better the differential effect by loss of FMRP. For example, the authors should show

the developmental change in the expression of specific subset of genes in both Wt and KO in

Figure S6d, e and f or DE along with significant p-value. The authors should already have

these data and could utilize the data fully to claim better. I believe people in FXS research

would be more interested in the changes happening in KO.

We thank the reviewer for the interest in the expression profile of FMRP bound mRNAs over

the developmental stages in the Fmr1-KO brain as we compiled and showed for WT in the

current FigS11a-b. Please note that the expression data of these genes in the WT mouse

brains were compiled from multiple published data sets (as cited in the main text), and the

corresponding Fmr1-KO data does not exist. Our intention with this analysis was to point out

potential avenues of discovery, but we do not make any claim for differences between

Fmr1-KO and WT. We agree that this would be interesting for researchers in the FXS field

and tried to highlight it by showing evidence for striking temporal patterns in the expression of

these genes. >> The explanation sounds clear.

8. ìThe cell type specific alteration of the transcriptome is a sensitive reflection of the cellular

status, and can serve as a first step towards an overview of the molecular impact of FXS.î If

this is the purpose of this manuscript, cell-type specific KD of Fmr1 including neuron,

endothelial cell and astrocytes should be shown. It is likely that the observed DE patterns of

this manuscript is combined effects of cell intrinsic and extrinsic functions of Fmr1 and/or

combined effects of multiple cells lacking Fmr1. Some experimental verification that can

support the authorsí conclusions should be presented. Otherwise, enthusiasm on this

manuscript would be minimal.

We thank the reviewer for affirming the significance of our data as a sensitive reflection of the

cellular status for the mouse model of FXS. The pathophysiology of the FXS is complicated

and involves the interaction of numerous cell types and pathways. Our data presents the first

of such with an overview, and we believe it will prove a valuable resource for the FXS field,

the field of neurodevelopmental diseases, as well as the field of neurodevelopmental biology

as a whole. And, as with the FXS disease, FMRP is lost in all cell types, the cellular status

reflected by the transcriptome changes could be due a combination of both intrinsic and

extrinsic disturbances. Indeed it would be an important next step to compare the

transcriptomic changes in cell type specific KO models, in order to parse out the cellular

intrinsic and extrinsic mechanisms. We agree that validation of our conclusions would

increase the impact of our study, however, given that this is the first study that dissects cell

type specific effects on mRNA levels in-vivo for the Fmr1-KO mouse, we believe that

validation work will be critical and we are sure that this study will motivate them.

Nevertheless, we thank the review for this insight and have added this point to the

Discussion. >> This was not addressed properly. At least, the single, most significant finding of their claims could be validated.

9. The arrangement of Supplemental figures is not easy to follow. The orders of

Supplemental figures should go in parallel with the main text.

We agree with the reviewer and have reordered the Supplemental figures. >> Well addressed.

10. Some of the figures and Supplemental figures were not cited in the main text, for

example, Figure S4a.

We thank the reviewer for pointing out this missing citation and have corrected it in the

current text. >> Well addressed.

11. There is no Figure 2f in Figure 2, which is cited in the main text

We thank the reviewer for pointing out this typo and have corrected it in the current text. >> Well addressed.

12. There is no reference (Utami et al, 2020 )cited in the Suppl figure legend. If the data was

from Utami, 2019, the cells must be hESC, not iPSC derived neuron.

We thank the reviewer for pointing out this citation was not up to date and have corrected it in

the current text. As for the correction from hiPSCs to hESCs, we have clarified the mentions

in our text, as both types of cell are present in the Utami et al. 2020 study, with the FMR1-KO

neurons being generated from hESCs and the human FXS neurons derived from iPSCs. >> Well addressed.

13. The comparison between mouse brain and human should be with human post mortem or

at least forebrain organoids at comparable developmental stage.

We agree with the reviewer that transcriptomic data from human FXS in vivo neurons, or

single cell data from post mortem cortex, would be ideal for comparison. However, to the best

of our knowledge, these data do not exist. However, in the past month (after our manuscript

had been reviewed), Kang et al. published the first ever human FXS organoid model (Kang et

al. 2021). Kang et al. present evidence for enhanced neuronal excitability through

immunocytochemistry, and in agreement with our results they report downregulated

neurodevelopmental pathway genes, as well as upregulated protein translation and

mitochondrial pathway genes by single cell RNA-seq, albeit without defining the cell types in

which such dysregulation happens. The conclusions by Kang et al. strongly corroborate our

findings. In return, our data complements that of Kang’s with unparalleled cell type resolution

(for example, as shown in Fig 4, the excitatory and inhibitory neurons exhibit a change in

opposite direction for translation and mitochondrial pathways, which Kang et al. mention only

as upregulated). We believe our data combined with studies such as these will provide

invaluable resources for the FXS research community. We have added the above to

Discussion. >> Citation should be added.

**Have all data underlying the figures and results presented in the manuscript been provided?**

Reviewer #1: Yes

Reviewer #2: Yes

Reviewer #3: Yes

PLOS authors have the option to publish the peer review history of their article (what does this mean?). If published, this will include your full peer review and any attached files.

Reviewer #1: No

Reviewer #2: **Yes: **Hansen Wang

Reviewer #3: No

---

## [Decision Letter · Decision Letter 2]

16 Mar 2022

Dear Dr Donnard,

Thank you very much for submitting your Research Article entitled 'Single Cell Transcriptomics Reveals Dysregulated Cellular and Molecular Networks in a Fragile X Syndrome Model' to PLOS Genetics.

The manuscript was fully evaluated at the editorial level and by independent peer reviewers. The reviewers appreciated the attention to an important topic but identified some concerns that we ask you address in a revised manuscript

We therefore ask you to modify the manuscript according to the review recommendations. Your revisions should address the specific points made by each reviewer.

[LINK]

Yours sincerely,

Mingyao Li

Associate Editor

PLOS Genetics

Gregory Barsh

Editor-in-Chief

PLOS Genetics

Reviewer's Responses to Questions

**Comments to the Authors:**

Reviewer #3: 1 and 8. This reviewer appreciates the authors' efforts, and understands that the authors tried to show meaningful change in trend of specific subset of genes by the lack of FMRP. but respectfully disagrees with the authors’ claims. As stated, the mild changes of multiple functionally related genes could contribute to the overall effect of FMRP loss in the pathogenesis of FXS. However, such arguments would get meaningful if the authors could provide experimental evidence to support the functional change/deficit caused by FMRP loss. Without such validations, enthusiasm on this manuscript is still moderate. The authors mentioned in their response and the reference (5) and (6), in the case where the average decrease per gene was only 20%, but could be validated by microarray and in vivo functional studies. Instead of working on single genes Mt1/Mt2 which were selected based on differential gene expression level, the authors could perform a knockdown of several functionally related genes to Mt1/Mt2, which is relevant to astrocyte-neuron communication based on GSEA, and then analyze if this could indeed mimic FMR1 KO phenotype in astrocyte-neuron communication in specific cell types. This way, the authors’ claim with GSEA could be fortified. It is not sure this manuscript without functional validation as is satisfies the PLOS Genetics criteria for publication in terms of substantial evidence for its conclusions.

13. As reviewer #1 pointed out that some of papers mentioned in the text or in the supplementary material do not appear in the final list, the authors should take a look at the missing references carefully. For example, newly added Kang et al. is missing in the reference list.

**Have all data underlying the figures and results presented in the manuscript been provided?**

Reviewer #3: Yes

PLOS authors have the option to publish the peer review history of their article (what does this mean?). If published, this will include your full peer review and any attached files.

Reviewer #3: No

---

## [Editor Report · Decision Letter 3]

27 Apr 2022

Dear Dr Donnard,

We are pleased to inform you that your manuscript entitled "Single Cell Transcriptomics Reveals Dysregulated Cellular and Molecular Networks in a Fragile X Syndrome Model" has been editorially accepted for publication in PLOS Genetics. Congratulations!

Yours sincerely,

Mingyao Li

Associate Editor

PLOS Genetics

Gregory Barsh

Editor-in-Chief

PLOS Genetics

Comments from the reviewers (if applicable):

**Data Deposition**

http://datadryad.org/submit?journalID=pgenetics&manu=PGENETICS-D-21-00817R3

**Press Queries**

---

## [Editor Report · Acceptance letter]

2 Jun 2022

PGENETICS-D-21-00817R3 

Single Cell Transcriptomics Reveals Dysregulated Cellular and Molecular Networks in a Fragile X Syndrome Model 

Dear Dr Donnard, 

We are pleased to inform you that your manuscript entitled "Single Cell Transcriptomics Reveals Dysregulated Cellular and Molecular Networks in a Fragile X Syndrome Model" has been formally accepted for publication in PLOS Genetics! Your manuscript is now with our production department and you will be notified of the publication date in due course.

With kind regards,

Zsofia Freund

PLOS Genetics

On behalf of:
